# Repetitive Behaviours and Restricted Interests in Individuals with Down Syndrome—One Way of Managing Their World?

**DOI:** 10.3390/brainsci7060066

**Published:** 2017-06-15

**Authors:** Sheila Glenn

**Affiliations:** School of Natural Sciences and Psychology, Liverpool John Moores University, Tom Reilly Building, Byrom Street, Liverpool L3 3AF, UK; s.m.glenn@ljmu.ac.uk

**Keywords:** Repetitive behavior, Down syndrome, Development

## Abstract

This paper argues that the repetitive behaviour and restrictive interests (RBRI) displayed by individuals with Down syndrome have mostly positive functions. However, as research has developed from interests in Obsessional Compulsive Disorder or Autistic Spectrum Disorder, unfortunately a view has arisen that RBRI in individuals with Down syndrome are also likely to be pathological. This is particularly the case in adults. The paper reviews: (a) measures employed and the perspectives that have been used; (b) the development in typically developing individuals, those with Down syndrome, and those with other conditions associated with intellectual disability; (c) positive and possible negative effects of RBRI; and (d) the need for more research. The conclusion is that, for their level of development, RBRI are helpful for most individuals with Down syndrome.

## 1. Introduction

This paper takes the view that much of the repetitive behaviour and restrictive interests (RBRI) displayed by individuals with Down syndrome are adaptive, and should not necessarily be seen as pathological. Previous work tends to have originated from clinical perspectives of, for example, obsessional and compulsive disorder (OCD) or autistic spectrum disorder (ASD). Evans, Leckman, Reznick, Henshaw, King, and Pauls [1] were particularly interested in the type of compulsive-like and routinized behaviours seen in young children, but which seem similar to behaviours seen in adults with OCD. Gesell, Ames and Ilig [2] had noted that at around 2 years of age children tend to become highly compulsive; they may be dependent on particular bedtime routines, expect things to be done “just right“, may have strong food preferences, and repetitive behaviour. They may want to hear the same story, are attached to a particular object, and repeatedly carry out the same play activities. If thwarted they are likely to react negatively. The ubiquity of these behaviours in early development suggests that they have an important function. Leonard, Goldberger, Rapoport et al. [3] argued that routinized and compulsive-like behaviours are important for reducing anxiety, for example in bedtime routines. Piaget [4] also emphasised the importance of repetition and “just right” behaviours in learning how to interact with the environment and developing classification skills. In other words, routines (and familiarity) for young children increase feelings of competence, reduce anxiety and free up cognitive capacity, all of which foster the likelihood that they will explore and learn new ideas and skills.

Recently there has been a debate about the need to define repetitive behaviour more precisely. For example Langen, Durston, Martien, van Engeland and Staal [5] noted that “repetitive behavior” has been used for behaviours linked by repetition, rigidity and invariance, as well as being seen as pathological in conditions such as OCD and ASD. The link with ASD is of interest, as repetition, rigidity and restricted interests are key features of autistic spectrum disorders, and are resistant to change. This may have led to the idea that similar (although less severe) behaviours in individuals with Down syndrome are also pathological. The present paper contests this view by considering the perspectives adopted in research, and the measures that reflect these.

## 2. Measures

Evans et al. [1] were among the first to provide a standardized measure of what they initially termed routinized and compulsive-like behavior for use with typically developing children. They developed the 19-item Childhood Routines Inventory (CRI), constructed from more user-friendly versions of DSM IV for compulsivity. Thus, it appears that one of their interests was whether these early behaviours could be related to later OCD. Factor analysis of the CRI produced two factors: First, “Just Right” (including items such as “arranges objects or performs certain behaviours until they seem just right”, “insists on certain belongings around the house being in their place”). Second, “Repetitive Behaviours” (including items such as “prefers the same household schedule or routine every day”, “acts out the same thing over and over in pretend play”). These two factors have been replicated in subsequent studies [6,7,8]. In addition, a few studies have found an additional factor of “Sensitivity to the environment” indexed by preference to certain articles of clothing (e.g., [6,8,9,10]). All studies have found internal consistency and test-retest reliability of the CRI to be high and significant. Evans et al. [1] also found a decline in CRI scores from five to six years of age. Many subsequent studies have used the CRI with typically developing children and in a growing number of studies with individuals with developmental delays, including Down syndrome.

In recent work, the Evans’ research group has employed the CRI as a measure of compulsive-like and routinized behaviours, but has changed description of the behaviours of interest to “Repetitive Behaviour and Restricted Interests (RBRI, [11]), noting that this term can comprise a wider range of behaviours such as compulsions, tics, stereotypies and self-injurious behaviour that may be seen in conditions such as ASD. Researchers studying the development of children with ASD have developed assessment tools with some overlap with the CRI. For example the Repetitive Behaviour Questionnaire [12] includes: (a) sensory and motor repetitive behaviours such as “rocking backward and forward or side to side”; having a “particular interest in smells of people or objects”; (b) restricted interests such as “playing the same music repeatedly”; insisting that “daily routines remain the same”. Clearly, the restricted items are similar to Evans et al. [1] CRI items; however, sensory and motor items do not appear on the CRI. Wolff, Boyd and Elison [13] have recently developed a further measure, arguing that previous scales were limited for the study of typically developing children, as there were significant floor effects for the latter. Their 34-item scale has both four factor and three factor solutions. The 4-factor solution produced factors named “Repetitive Motor”, “Ritual and Routine”, “Restricted”, and “Self-Directed Behaviour”. They suggest that the 4-factor solution may be of more use with children with neurodevelopmental disorders, particularly in relation to self-injurious behaviour. Their sample needs to be extended to older age groups, but the “Ritual and Routine” and “Restricted” factors are similar to the CRI scale. 

In 2017 Evans, Uljarevíci, Lusk, Loth and Frazier [14] reported the development of a revised CRI measure for children (CRI-R), together with a 55 item Adult Routines Inventory. There was a representative sample of 3032 children aged from 12 months to 17 years 11 months; 26% of the children had neurodevelopmental disorders. The scale has very high reliability; and a factor analysis produced two factors: (a) “rigidity and insistence on sameness”; (b) “repetitive sensory and motor behaviour and compulsions”. Clearly, this scale will be useful for future wide-ranging studies in developmental and clinical research.

Given that there is more work using the original CRI with typically developing children, and research with individuals with Down syndrome, the present review is based on the Evans et al. [1] development of the CRI. It provides a typical developmental framework, within which studies with individuals with Down syndrome can be set. It covers compulsive-like and repetitive behaviours, as well as the routines seen in young children. Where appropriate, alternative assessments are referenced.

Several questions arise in relation to research with individuals with Down syndrome:
What is the typical developmental trajectory of RBRI?What is the developmental trajectory in those with Down syndrome?Are RBRI specific to Down syndrome?What are potential positive functions of RBRI?What are potential negative consequences of RBRI?


### 2.1. What Is the Typical Developmental Trajectory of RBRI?

Research with typical samples is important as it provides a normative developmental framework. 

Using the CRI, Evans et al. [1] collected data on 1488 typical children aged from 8 months to 6 years. They found that the intensity of RBRIs increased around 2 to 4 years and declined significantly in the 5 to 6-year old age group. However, Glenn and Cunningham [15] found no significant decline with increasing verbal mental age (VMA) or chronological age (CA) in children with mean VMAs up to 7 years and mean CA of 5.3 years. Tregay et al. [16] found no significant decline in children aged 3–9 years. Zohar and Bruno [17] examined the level of obsessive-compulsive behaviours in children aged 8 to 14 years using the Maudsley Obsessive Compulsive Inventory [18], and found a significant decrease in such behaviours from eight to 14 years. Evans et al. [14] also found no decline in the Rigidity and Sameness factor until after the 8–13 years age group; however, the results did include 26% of children with neurodevelopmental disorders. Overall such results imply that the proposed decline of similar behaviours in typically developing children might be later than the 5–6 years suggested by Evans et al. [1]. 

Glenn et al. [6] replicated the Evans et al. study with 1369 children aged 2–11 years, also including a measure of OCD. Forty one children with developmental disabilities were excluded from the analysis. A significant decline in RBRIs started between 6 and 8 years of age; a minority of children continued to have high levels up to 11 years of age. There were significant correlations between CRI and OCD scales, and children designated at risk for OCD had higher CRI scores. Therefore, the question arose as to whether such behaviours served an adaptive function or became associated with pathology. Only one child had a diagnosis of OCD following a streptococcal throat infection (PANDAS [19]). None of the other parents described their children’s repetitive and compulsive like behaviour as problematic or interfering with their lives. Childhood worries reported by parents increased significantly from 7 years of age and for these older age groups there were consistent relationships between ratings of frequency of worries and intensity and number of RBRI. These results confirmed previous suggestions that one of the functions of RBRI may be to reduce anxiety [20,21,22,23].

Glenn and Nananidou [22] followed up 109 children, aged 8 to 18 years, from the Glenn et al. [6] study, 6 years later. They also measured OCD behaviours, worries and fears. There were no significant differences on Time 1 (2012) CRI measures between participants and the original sample. CRI scores decreased over the 6-year period. Dividing the group into those with and without problems or difficulties (reported by their parents), found that those without difficulties decreased their CRI scores significantly whereas those with difficulties did not. Difficulties (none of which had been reported at time 1) included learning difficulties, ASD, Attention deficit hyperactivity disorder, and depression, all previously reported to have associations with OCD or repetitive behaviour (e.g., [24,25,26]). In contrast to CRI scores, OCD behaviours increased significantly, however, only one child in the sample had an OCD diagnosis. There were significant correlations between CRI, OCD and fear and worry scores. Therefore, RBRI were still present in older children, were associated with worries and fears, but there was only one OCD diagnosis. It is possible that if in the future, the behaviours started to interfere with everyday life, a psychiatric diagnosis would result. However, at the time of the study, these behaviours were not pathological. 

Thus, although there is very little longitudinal research, there is some evidence of RBRI in older typically developing children with no clinical diagnoses i.e. they are part of typical development.

### 2.2. What Is the Developmental Trajectory in Individuals with Down Syndrome?

Evans and Gray (2000) [27] matched typically developing children for mental age (MA) with individuals with Down syndrome. They found that those with Down syndrome showed similar declines in CRI scores to typically developing children. For typically developing children with MAs over 5 years, there were positive correlations between CRI scores and adaptive functioning. In contrast, for both the younger and older MA children with Down syndrome, there were positive correlations between total CRI scores and problem behaviours, assessed on the Child Behavior Checklist (CBCL [28]). The number of CRI items endorsed was similar, suggesting developmental typicality, but the behaviours were rated as significantly more intense for the children with Down syndrome. Glenn and Cunningham [6] replicated the study, also including adults with DS. However, they found no decreases with age up to 6 years either in typically developing children, or in those with Down syndrome up to verbal mental age (VMA) 6 years. Nor were there decreases with increasing VMA in adults with Down syndrome. For children with Down syndrome with VMAs greater than 5 years and all adults with Down syndrome, CRI scores were associated with behaviour problems. This association was also found by Evans, Kleinpeter, Slane and Boomer [11], who also noted little decrease, and also some increase in “just right” behaviours in their older group with Down syndrome. With respect to behavior problems, Cunningham and Glenn [29] re-analysed their 2007 data and concluded that the raised rating scores seen on the CBCL were predominantly caused by items related to intellectual disability e.g., “acts young for age”; “can’t concentrate”; “too dependent”; “difficulty in learning”; “clumsy”; “speech problems”). Removing items relating to intellectual disability, they found that CRI scores were only significantly associated with the worry item of the CBCL. This further supported the suggestion that one of the functions of RBRI may be to reduce anxiety.

In relation to the question of pathology of RBRI, Glenn et al. [8] studied 125 adults with Down syndrome aged 18 to 43 years assessed for mental health problems (Prasher, Glenn, Cunningham, Glenholmes, Arshad, and Kirby [30]). Parents or carers completed the CRI, and the Strengths and Difficulties Questionnaire [31] in order to get measures of worries and fears. Mean VMA (*n* = 91) was 4 years 5 months, range 12 months to 13 years 6 months. Compared to typical children, there were higher scores on the CRI measure. There were no significant correlations with either CA or VMA, confirming previous results (Glenn and Cunningham [6]). There were higher CRI scores for those with a psychiatric diagnosis, but this was not significant. There were significantly higher levels of worries and fears in the psychiatric group, and a logistic regression analysis showed that the fear score was the significant predictor of psychiatric diagnosis, not the CRI score. Uljarevic and Evans [10] also found significant associations between CRI scores and fears in 38 children with Down syndrome, mean chronological age (A) 10 years 5 months, SD 3 years 10 months, mean mental age (MA) 4 years 4 months. These results again suggest that RBRI have an adaptive function in relation to anxiety reduction. 

### 2.3. Are RBRI Specific to Down Syndrome, or to Intellectual Disability

The question then arises about whether the raised level of RBRI seen in individuals with Down syndrome is specific to that condition or also a feature of those with other developmental disabilities. Higher levels of RBRI compared to typical are found in many other conditions associated with developmental disabilities. This finding is most pronounced in autism spectrum disorder (ASD), and is one of the defining characteristics of that condition. Uljarevic and Evans [10] found significantly higher CRI scores in 41 children with ASD than in those with Down syndrome. Similarly, Glenn and Egan [32] compared CRI scores of 12 children with ASD matched for CA with 36 with learning disabilities; the children with ASD had significantly higher ratings from both teachers and parents. Those with ASD also had significantly higher scores than 40 CA matched children with Down syndrome.

There have also been studies with children with other conditions associated with intellectual disability. For example, Wigren and Hansen [33] found significantly more intense Repetitive behaviour on the CRI for 50 individuals with Prader-Willi syndrome (aged 5–18 years), compared to 50 typically developing children aged 4 years. Greaves, Prince, Evans and Charman [34] matched children with Prader-Willi syndrome and children with autism for age and found similar high scores. James, Riby and Rodgers [35] found parental reports of restricted and repetitive behaviours in children with Williams syndrome. Moss, Oliver, Arron, Burbridge and Berg [36] found increased levels of repetitive behavior, using the Repetitive Behaviour Questionnaire [37] in individuals with Prader-Willi syndrome (*n* = 189), Fragile X syndrome (*n* = 191) and Cri-du-Chat syndrome (*n* = 58). The mean age was 18.5 years (SD 10.3). A review by Leekam, Prior and Uljarevic [38] noted little evidence on change with age in restricted and repetitive behaviours in childen with autism. They highlighted the need to study the adaptive functions of such behaviours. In relation to possible functions, Rodgers, Riby, Janes, Connolly and McConachie [39] demonstrated a positive association between anxiety and levels of repetitive behaviours in children with autism. Joosten, Bundy and Einfeld [40] found that children with autism were more likely to display repetitive behaviours in an educational setting when children had finished a task or when they had to start a new task. They also showed more signs of anxiety in such situations. This suggests that the children coped with anxiety by resorting to repetitive behaviours and this may not be advantageous for learning new skills.

In summary there is much evidence indicating that a variety of conditions associated with intellectual/developmental disabilities have increased levels of RBRI compared to typically developing children; they are not unique to Down syndrome.

### 2.4. Potential Positive Functions of RBRI

#### 2.4.1. Executive Function

Executive functions (EF) are a set of cognitive processes—including attentional control, inhibitory control, working memory, and cognitive flexibility, necessary for the cognitive control of behavior [41]. They are necessary for adaptation to new circumstances, rather than for well-learned behaviours or familiar routines. Researchers have suggested that as executive function develops so the need for RBRI decline. For example, Tregay et al. [15] found significant relationships between deficits on some EF tasks and higher levels of RBRI in typically developing children aged 5 to 9 years. Pietrefesa and Evans [21] also found similar relationships in children aged 6 to 8 years, but not on the same EF tasks. 

Working memory is an important aspect of EF, and is a particular difficulty for individuals with Down syndrome [42], so it has been suggested that there are likely to be EF deficits in individuals with Down syndrome. Lanfranchi, Jerman, Dal Pont, Alberti and Vianello [43] found impairments in several cognitive processes (including set shifting, planning, problem solving, working memory and inhibition/perseveration) in 15 adolescents with Down syndrome, and concluded that EF deficit is a characteristic of Down syndrome. Similarly, Carney, Brown and Henry [44] found deficits in working memory and set shifting in 25 children and adolescents with Down syndrome matched for MA with typically developing children. Memisevic and Sinanovic [45] studied EF in children with intellectual disability, including 30 children with Down syndrome (mean age 11.8 years, SD 2.8). They assessed Inhibition, Shifting, Emotional control, Initiating, Working memory, Planning, Organising material and Monitoring. All children showed an EF deficit compared to typically developing children, the only significant difference for aetiology was that children with Down syndrome were significantly worse at Shifting. Such results suggest that EF deficits are not unique to Down syndrome but rather are a function of intellectual disability. 

In summary, there is a need for more research on EF deficits, particularly in relation to developmental age, and the possible developmental functions of RBRI in either supporting or hindering EF in individuals with Down syndrome.

#### 2.4.2. Reduction of Anxiety

As described previously there is considerable research supporting the original view of Leonard et al. [3] that RBRI are important for reducing anxiety or fear. This has been found in typical children [8,20,21,22,23,46]. 

A few studies have demonstrated similar findings with individuals with Down syndrome [8,10,28]. There are similar results for children with ASD [35,36]. 

However, these are correlation studies, and an anonymous reviewer pointed out that although RBRI may reduce anxiety or fear, this may not be an adaptive strategy in the long term for dealing with anxiety provoking situations. The development of executive function skills may be more important.

#### 2.4.3. Adaptive Behaviour

Studies have looked at relationships between RBRI and adaptive behaviour as measured by the Vineland Scales of Daily Living. Glenn and Cunningham [14] found that for both typically developing children and those with Down syndrome with MAs of 5 years or less, there were significant positive correlations between CRI measures and total Vineland Screener scores [47]. For older typically developing children, children with Down syndrome, and adults there were no significant correlations with Vineland scores i.e. there were no negative effects of RBRI on daily living skills for older individuals.

In contrast, Evans et al. [11] carried out a longitudinal study on RBRI and adaptive behaviour over a 2-year period. They found that for typically developing children, there were significant positive correlations between CRI scores at time 1 (2–4 years) and Vineland scores at time 2 (5–11 years). However, for those with Down syndrome matched for MA, there were negative correlations between CRI scores at time 1 and Vineland scores at time 2. They argued that for typically developing children, RBRI in early development serve an adaptive function in later development; also that “This is rather in contrast to the belief that early RBRI are a risk factor for later symptoms of OCD or ASD” ([11] p. 7). They suggested that for those with Down syndrome, persisting RBRI might interfere with the teaching of daily living skills. 

Larkin et al. [48] carried out a longitudinal study with typically developing children from 26 to 61 months of age, using the Repetitive Behaviour Questionnaire. There was a negative relationship between sensory and motor repetitive behaviour at 26 months and later language performance, and theory of mind. However, there was no relationship between Rigidity/routines/restricted (RRR) interests at 26 months and later performance on any measures. One limitation of the study was that there was no assessment of RRR at later ages. It is also important to note that one emphasis of this group was in possible prediction of ASD from early measures.

Thus, there are mixed results from the few studies outlined above. It is possible that RBRI have different functions from the processes required to learn daily living skills; the latter require individuals to learn social norms. However, it is important to ask whether RBRI (largely self-generated) interfere with the acquisition of daily living skills. An observational study looking at the education or skill training of children and adults with Down syndrome would give some indication of whether RBRI do have inhibiting effects on the learning of daily skills. Joosten et al. [36] showed that children with ASD showed more anxiety and more repetitive behaviours when presented with an unfamiliar task in school. It would be informative to carry out such studies with individuals with Down syndrome in natural, as well as experimental, settings. 

### 2.5. Possible Negative Consequences

As outlined above the results of Evans et al. [11] highlighted a potential negative effect of RBRI on adaptive behaviour. If RBRI start to interfere with engagement with the individual’s environment, it would be important to examine environmental factors that might have caused this increase, as well as emotional difficulties. In addition, the role of problems with EF deserve more research. All these factors are important to consider when initiating intervention. OCD is not a common diagnosis in individuals with Down syndrome [49]. However, a dual diagnosis of Down syndrome and ASD is becoming more common. For example Richards, Jones, Groves, Moss and Oliver [50] in a recent meta-analysis, reported a pooled prevalence rate of 16% of ASD diagnosis. This is likely to be associated with a particularly high rate of RBRI, and such findings highlight the need to be aware of possible ASD in individuals with Down syndrome.

## 3. Conclusions

First, we should note that restricted behaviour and repetitive interests tend to diminish in typically developing children at around 7 years of age. However, some teenagers with no psychiatric diagnosis still have RBRI. 

The average mental age of adults with DS is around 5–6years, so it is probable that the relatively high level of RBRI are still adaptive at this developmental age. Some higher functioning individuals may progress beyond this, although the problems with working memory and executive function may hinder some aspects of development. There is also evidence that RBRI are not unique to Down syndrome, but are also found in a variety of other conditions associated with intellectual disability. 

There is some indication that RBRI may interfere with the development of daily living skills. Further research is need to confirm this result. More detailed observational rather than questionnaire studies would inform this issue. The CRI-R developed by Evans et al. [14] will be very useful for typical and clinical studies. Also Honey, McConachie, Randle, Shearer and Le Couteur [51] have suggested additional measures such as asking both parents and youngsters with Down syndrome about RBRI, if they are problematic, and what strategies can be used to widen interests. 

However, for most individuals with Down syndrome RBRI are developmentally appropriate; the behaviours will only become problematic if they increase in frequency to the extent that they are interfering with learning other skills. 

In addition, it is important to be clear about the definition of behaviours. For example, there is a distinction between repetition (necessary for learning) and actions that appear repetitive and not leading to new skills. The issue about some of these repetitions (e.g., repeated play with the same well-mastered jigsaw, insistence on hearing/reading the same story or viewing the same video) is that observers may not appreciate how much information is still being learned or experienced as novel by the child or individual. Piaget [52] gave an account of how the same theme of sliding down a slide, actually involved many different variations as children learned about their environment. If the term “repetitive behaviours and restricted interests” were to be replaced with a more neutral term, this would remove some of the negative connotations of the former.

There are also the feelings of security and sense of mastery associated with the familiar. Faced with uncertainty and associated anxiety, we all tend to revert to familiar knowledge or actions. We establish familiar routines to cope with a complex world. People with intellectual disabilities are no different from the rest of us.

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
