# Peer review of "Repetitive Behaviours and Restricted Interests in Individuals with Down Syndrome—One Way of Managing Their World?"

_brainsci, 2017, doi:10.3390/brainsci7060066_

Round 1

Reviewer 1 Report

This is a well-written and well-organized review of research on the repetitive behaviors and restricted interests of individuals with Down syndrome. The author focuses her review largely on the development of RBRI in studies using the Childhood Routines Inventory, and subsequent replications studies.

Early in the review the author correctly identifies the limitations of the CRI in its relative lack of sensory items. However, it must be noted that the CRI-Revised was recently published as a greatly expanded measure that includes compulsions, tics, stereotypies, sensory sensitivities, as well as those repetitive behaviors that present throughout typical development. See http://www.jaacap.com/article/S0890-8567(16)31890-1/fulltext?rss=yes. This recent paper is also noteworthy because it uses a nationally-representative cohort (the first study on a representative population/community cohort on RBRI to date) across a large age range spanning 1 to 18 years of age. It demonstrates that the ages-related changes reported in earlier work, but also suggests slightly later decline. This somewhat resolves some of the inconsistencies between the earlier reports and some of the replication studies. I believe it should be included in the current review.

On pages 3 and 4 the authors review the Evans and Gray (2000) and the Evans, Klinepeter, Slane & Boomer (2014). However, they do not (unless I missed it) appear to indicate that the 2014 paper noted a leveling-off, and even some increase in “just right” behavior in the older DS cohort. That they did not find the decreases noted in earlier papers would seem to be noteworthy, as is a departure from earlier reports, and presumably consistent with the replication work of Glenn and her colleagues.

The author writes: “Uljarevic and Evans10 163 also found significant associations between CRI scores and fears in 38 164 children with Down syndrome, mean CA 10 years 5 months, SD 3 years 10 months, mean MA 4 years 165 4 months. These results again suggest that RBRI have an adaptive function in relation to anxiety 166 reduction.” This is interesting issue: whether a correlation between RBRI and fear or worry means that RBRI play an adaptive role in regulating fear and worry, or whether it represents a failure to do so (I recall an early paper by Passman entitled: “are children with security blankets insecure?”). It is an open question still, and may very well be that executive function determines in part, whether RBRI are effective at reducing fear and worry. This issue could be expanded upon somewhat.

A minor point: Reference 34, the lead author’s name is spelled “Leekam” not “Leekham”

Overall this is a very nice summary of the work on RBRI in DS. I offer the comments for the author’s consideration.

Author Response

Thank you for the helpful review. I was unaware of the latest version of the CRI-R; I have added this to the re4view - I think it will be very useful in future research.

I have also added the results from Evans et al. 2014 as you suggested.

I found your question as to the role of RBRI in reducing anxiety, thought provoking and have added a little about the need for more research on this question.

Reviewer 2 Report

Thank you for the opportunity to review the manuscript "Repetitive behaviours and restricted interests in individuals with Down syndrome – one way of managing their world?" The topic is of interest to me and I do feel with some revision, the paper offers an interesting theoretical perspective on RRBs in DS. In it's current form, the paper left me wanting more information about the topic as the scope is rather limited.

When writing a theoretical/review piece, the author should offer some recommendations/points for discussion. Are RRBs of use within intervention as they can be in ASD? Is there any work involving self advocates - this is a big move in ASD research to understand the meaning of RRBs to individuals with ASD. Are there any studies that measure change with intervention? Highlighting the future directions of the research area would give the paper more significance. 

More detail about the types of behaviours children/adults with DS demonstrate compared to ASD/OCD would have been appreciated. Are the profiles of behaviour similar or different?

By limiting the review to mainly studies that used the CRI significantly limits the scope of the paper and is seen as a major limitation. There are other relevant studies (Honey et al., for example) that used other measures (RBQ, RBS-R) that would also be of importance.

I would argue that the RBS-R is the most widely used tool for this area. The Wolff et al paper is based on the RBS-R NOT the RBQ. This needs correcting. 

Author Response

Thank you for your helpful comments.

I have added a little more on intervention, and added the Honey et al. reference. I

The only intervention to reduce RBRI  when we were studying this issue was a few young people who had been prescribed medication to stop repetitive behaviour. It seemed to us that this  behaviour was just a more intense version of what we were seeing in typically developing children of the same MA, but was being seen as pathological by some.  This experience led  to us highlighting the typical nature of much RBRI in those with Down syndrome.

For this reason we restricted the review to the CRI, developed on typical children. A new measure developed by Evans et al (2017) J Amer Acad Child Adolesc Psychiat extends the CRI for large representative samples  both children and adults and should be v useful for future studies in for ex profiling DS/Autism differences.

Reviewer 3 Report

Brief summary

The aim of this article was to argue that repetitive behaviour and restrictive interests (RBRI) have positive functions for individuals with Down syndrome. The paper provides a review of research investigating RBRI in both typically developing populations and individuals with Down syndrome.

Broad comments

The title of the manuscript indicates that the article will focus on RBRI in Down syndrome but there is considerable focus on RBRI in typically developing populations throughout the manuscript. There is a distinct lack of evidence relating to RBRI in Down syndrome and I therefore have concerns that the aim of the article is not addressed. In order to improve the manuscript, I recommend that either the title is changed to match the content within the manuscript, or the content of the manuscript needs to be revised so that there is a greater focus on research that has investigated RBRI in individuals with Down syndrome.  

The ‘measures’ section (p. 2) could be more concise as this is not particularly relevant to the aim of the paper.

It is not clear whether the ‘potential positive functions of RBRI’ section (p. 5) is related to just Down syndrome or whether this section is also focusing on typically developing populations. As the aim of the paper is to argue for the positive functions of RBRI for individuals with Down syndrome, I was expecting this section to focus on Down syndrome. However, much of the research that is presented in this section relates to typical development. In addition, it is not clear how this section of the manuscript is relevant to the aim of the paper as there is minimal evidence presented to suggest that RBRI have positive functions for individuals with Down syndrome. In particular, the ‘executive function’ section (p. 5) does not provide any evidence to support the view that RBRI have a positive function for executive function. Therefore, this section of the paper could be revised so that the positive functions of RBRI for individuals with Down syndrome are clear and appropriate evidence is used to support this argument.

In line with the aim of the paper, the concluding paragraph could explain how the paper has demonstrated that RBRI have positive functions for individuals with Down syndrome. At present, it is not clear how the arguments that the author has presented address the aim of the paper.

The paper provides an overview of research that has used the Childhood Routines Inventory (CRI) to assess RBRI in both typically developing populations and individuals with Down syndrome. In my view, the manuscript could be improved by including research that has investigated RBRI using other measures, such as the Repetitive Behaviour Questionnaire (RBQ). This measure is mentioned in the ‘measures’ section of the paper but within the manuscript, only one study is discussed which has used the RBQ to assess RBRI. Reference to studies using a range of measures of RBRI would improve the validity of this manuscript.

Specific comments

Line 28: OCD does not need to be in brackets.

Line 46: “by considering the” appears consecutively twice within the sentence.

Line 64: “In recent work, the Evans’ research group has employed the CRI as a measure”. Need to clarify what the CRI has been used to measure.

Line 74: state what the “further measure” is.

Line 97: state the chronological age of the children in the Glenn & Cunningham (2007) study.

Line 98: clarify what VMA and CA refer to.

Lines 109 – 110: the author states that one of the participants received a diagnosis of OCD. Were any other diagnoses (e.g. anxiety, autism spectrum disorder, ADHD) reported for any of the participants in this study?

Line 118: who are the “non-participants”?

Lines 153 – 155: there are two different references for the same study.

Lines 170 – 171: could you provide some examples of other conditions associated with developmental disabilities in which individuals have high levels of RBRI?

Lines 187 – 189: need to explain why the Joosten, Bundy & Einfield (2012) study is relevant.

Lines 198 – 199: could you provide a reference for the statement that “researchers have suggested that as executive function develops so the need for RBRI decline”?

Line 227: clarify what RCB refers to.

Lines 228 – 232: explain the significance and importance of the findings from the Glenn & Cunningham (2007) study in relation to RBRI and adaptive functioning.

Lines 239 – 240: evidence is presented which suggests that “persisting RBRI might interfere with the teaching of daily living skills” for individuals with Down syndrome. If this is the case, I am uncertain as to why this is presented in the ‘potential positive functions of RBRI’ section of the manuscript as this appears to be a negative function of RBRI for individuals with Down syndrome.

Lines 270 – 271: is there any evidence to support the claim that high level of RBRI are adaptive for adults with Down syndrome?

Lines 278 – 280: is there any evidence to support the claim that “for most individuals with Down syndrome RBRI are not problematic”?

Within the text, from reference 29 onwards, the in-text citations are numbered incorrectly. These need to be updated.

Author Response

Thank you for your comments.

I hope that I have addressed all your specific comments, some comments provided by other reviewers have also been added to this section.

With respect to the broad comments;

Our view is that the development of individuals with Down syndrome should be set within the context of typical development, and that much of the behaviour of this group is developmentally appropriate and should not be seen as pathological. This is the reason for  a review of the more extensive (but not always consistent) work with typical children groups. The new measure developed by Evans et al (2017) - not available when this review was first written-  has now been added and will be v useful as it enables RBRI in clinical populations to be seen within a framework of typical development.

I have added more discussion on +ve and -ve functions of RBRI.

Measures such as the RBQ are geared towards autism and ASD, have floor effects for many children, and the new CRI-R referenced above is, in my opinion, more likely to provide a wider view of children and adult behaviour.

Round 2

Reviewer 3 Report

The author has made the focus of the article clearer and this has improved the structure and coherence of the manuscript.

Minor Specific Comments

Line 25: OCD should be in brackets here.

Line 28: OCD does not need to be in brackets here.

Line 44: the end of the sentence should appear as ‘precisely’.

Line 79: what is the RBS-R?

Line 111: use the term ‘framework’ instead of ‘frame’.

Lines 238 – 239: the conclusion of this section is that “there is much evidence indicating that a variety of conditions associated with intellectual/developmental disabilities have increased levels of RBRI”. More examples of these conditions could be provided in this section as there are currently only two examples (ASD and Prader-Willi syndrome).

Line 255: “found deficits in in working”. One of the “in” needs to be removed.

Line 300: the term “sensory” does not need a capital letter.

Author Response

I have changed the specific comments and added more references on repetitive behaviour in individuals with genetic syndromes associated with intellectual disability.